# A Labeling Method for Financial Time Series Prediction Based on Trends

**DOI:** 10.3390/e22101162

**Published:** 2020-10-15

**Authors:** Dingming Wu, Xiaolong Wang, Jingyong Su, Buzhou Tang, Shaocong Wu

**Affiliations:** The College of Computer Science and Technology, Harbin Institute of Technology, Shenzhen 518055, China; hitsz_wudingming@outlook.com (D.W.); sujingyong@hit.edu.cn (J.S.); tangbuzhou@hit.edu.cn (B.T.); wushaocong2013@gmail.com (S.W.)

**Keywords:** financial time series, stock prediction, machine learning, labeling method, deep learning

## Abstract

Time series prediction has been widely applied to the finance industry in applications such as stock market price and commodity price forecasting. Machine learning methods have been widely used in financial time series prediction in recent years. How to label financial time series data to determine the prediction accuracy of machine learning models and subsequently determine final investment returns is a hot topic. Existing labeling methods of financial time series mainly label data by comparing the current data with those of a short time period in the future. However, financial time series data are typically non-linear with obvious short-term randomness. Therefore, these labeling methods have not captured the continuous trend features of financial time series data, leading to a difference between their labeling results and real market trends. In this paper, a new labeling method called “continuous trend labeling” is proposed to address the above problem. In the feature preprocessing stage, this paper proposed a new method that can avoid the problem of look-ahead bias in traditional data standardization or normalization processes. Then, a detailed logical explanation was given, the definition of continuous trend labeling was proposed and also an automatic labeling algorithm was given to extract the continuous trend features of financial time series data. Experiments on the Shanghai Composite Index and Shenzhen Component Index and some stocks of China showed that our labeling method is a much better state-of-the-art labeling method in terms of classification accuracy and some other classification evaluation metrics. The results of the paper also proved that deep learning models such as LSTM and GRU are more suitable for dealing with the prediction of financial time series data.

## 1. Introduction

A time series is a set of observations, each one being recorded at a specific time [1]. Prediction of time series data is a relatively complex task. Since there are many factors affecting time series data, it is difficult to predict the trend of time series data accurately. Time series forecasting aims at solving various problems, specifically in the financial field [2]. In the course of financial market development, a large number of studies have shown that the market is non-linear and chaotic [3,4,5], especially for the financial time series data such as the values of stocks, foreign exchanges, and commodities in the financial market that are sensitive to external impact and tend to fluctuate violently. Such time series data often have strong non-linear characteristics [6]. How to better predict the trend of financial market is of great significance for reducing investment risk and making financial decisions.

There have been a lot of studies on the prediction of financial market. In particular, we focus on the efficient market hypothesis [7]. There are many studies on market efficiency [8]. Sigaki et al. used permutation entropy and statistical complexity over sliding time-windows of price log returns to quantify the dynamic market efficiency [9]. Some studies have pointed out that China’s stock market is relatively ineffective [10,11]. Then it is feasible to forecast the ineffective market by using China stock transaction data.

Traditional methods of financial market prediction analysis mainly include technical analysis and fundamental analysis. Technical analysis refers to the methods of making all financial trading decisions of stocks and derivatives by taking market behaviors as the object to judge market direction. Specifically, in the financial market, researchers believe that price and volume data can be used as indicators of future price changes, and hence can provide important information to promote returns [12]. Fundamental analysis, also known as basic analysis, is based on the intrinsic value of securities. When combined with industry and economic data, fundamental analysis focuses on the analysis of various factors affecting the price and trends of securities to determine the intrinsic value of companies and identify securities with mispriced prices, so as to determine potential securities for investment [13]. Also, there are some time series studies on news and event driven factors [14].

With the rise of artificial intelligence, more and more disciplines are being transformed by artificial intelligence, including much analysis and decision-making work in the financial field. How to use artificial intelligence technology to accurately predict stocks on historical data is a new direction of artificial intelligence technology. At present, machine learning models are increasingly used in the analysis of financial time series data [15], cluster analysis [16], classification research and regression analysis, especially the regression analysis of return [17], as well as hybrid models forecasting, and financial market liquidity research [18]. Behavioral finance focuses on the relationship between transaction behavior and price, etc. [19]. Wasik et al. studied a model that combined neural networks and high-frequency trading [20,21]. How to evaluate the prediction performance of a machine learning model in the financial market is also a hot topic. The volatility of the stock market is not consistent, and there are periods of low and high volatility, such as a period when a financial crisis occurs. The prediction performance of the model in alternative periods is different, and even in mature financial markets and emerging markets, the forecast effects are also different [22]. According to Cerqueira et al.’s study, it was concluded that cross validation is usually the most effective method to evaluate the performance of the model [23]. Siriopoulos et al. studied the performance of artificial neural networks in stock investment [24], and also analyzed the application of artificial neural networks and emerging markets in 1995 [25]. Samitas et al. studied a system based on structured financial networks named “early warning systems” which was used for warning of the contagion risks, and excellent results were obtained, which is of guiding significance for the performance of machine learning models in times of economic crisis [26]. Galicia et al. proposed ensemble models for forecasting big data time series with decision trees, gradient boosted trees and random forest, and the prediction result was superior to the single model [27]. Valencia et al. studied the price change by combination with sentiment analysis and machine learning [28]. Studies on the combination of machine learning and financial markets have been diversified, and stock price prediction is a hot field. A variety of solutions have been developed and applied [29,30,31].

The process of applying machine learning to financial trend prediction mainly depends on the models established after processing relevant historical data characteristics, which are used with the matching model to forecast market after relevant patterns are learnt. The key point is to identify the important features and only use them as input, and not all features can improve the prediction accuracy of the prediction model [32]. Sometimes, the features are highly correlated, and then data dimensionality reduction is required, which refers to the PCA algorithm [33], or the data contain too much noise and the data needs to be denoised, which refers to the wavelet transform algorithm [34]. At present, two kinds of expression methods are available for the features of time series data. One is based on the statistical characteristics of time series data, and the other is based on the entropy features. The entropy characteristics of time series are usually covered in the following three categories: binned entropy [35], approximate entropy [36] and sample entropy [37]. Miśkiewicz proposed a entropy correlation distance based method (ECDM), which was applied to several time series, and the results proved that the entropy correlation distance measure is more suitable in detecting significant changes of the time series [38], which can also be used to test some economy globalization processes [39]. Machine learning can better quantify various characteristics, make quantitative decisions, and respond quickly to market information and can predict trend changes more quickly, thereby guiding decision makers to conduct corresponding risk control and make investment decisions. In particular, the deep learning developed in recent years has the characteristics of unsupervised learning ability [40]. In order to predict time series data, a large number of researchers have used technical analysis, basic analysis, time series analysis, machine learning and other techniques to predict the future trend of financial markets [41].

The prediction methods of time series data used in machine learning mainly include regression and classification, among which the regression algorithm is used to predict the absolute value of rising and falling prices. In the progress of model training, it is not necessary to label the data, although the classification algorithm often needs to label the data for the training of classification models.

In order to facilitate the comparison of results, this paper reviewed the literature on regression analysis and classification research carried out for the prediction of time series data. Regression analysis has been widely used in time series analysis and prediction. Chen. et al. used trading data to predict closing prices on the next day, the next 3 days, the next 5 days, and the next 10 days, etc. Their evaluation metrics were MAPE and RMSE [42]. Talavera-Llames et al. introduced a novel algorithm for big data time series forecasting. Its main novelty lies in its ability to deal with multivariate data with the evaluation metrics of MRE and MMRE [43]. There are many studies on the regression analysis and prediction of the market including but not limited to the studies described in [44,45,46,47,48,49].

It becomes a classification problem of numerical prediction if the goal is to predict the future direction of rising and falling time series data. Zhang et al. used monthly data to forecast the return rate of the following month by establishing a classification model [50]. The research of Leung et al. showed that the classification model is superior to level estimation models in predicting the direction of the stock market and maximizing the return on investment transactions. It is believed that better predictions can be made by using a classification algorithm to build a machine model. A previously proposed data labeling method is to label the data by comparing the logarithmic rate of return on the next day with the logarithmic rate of return on the present day [51]. Chong et al. used classification algorithms to study the rising and falling market, and it was indicated that the problem of up or down prediction is essentially a two-class classification problem, in which *y*∈{−1,1} denotes the class label and *y* = 1(or −1) represents the up (or down) movement of the market rate of return on the next day [52]. Wang et al. studied the opening price of the stock market on the next day. In their opinion, the positive or negative direction of the opening return rather than the absolute value of the opening return itself is of the foremost interest in reality because it can provide advice on the direction of trading. The authors used zero as the cut-off point and transformed *y*∈{0,1} (which denotes the label) into a binary variable [53]. Mojtaba Nabipour et al. trained four groups of machine learning models by comparing the stock data at 1, 2, 5, 10, 15, 20, and 30 days in advance [54]. Other studies on market direction prediction basically used the same approach by comparing the price or yield rate at two different time points [55,56,57,58,59].

It can be concluded from the traditional research that the idea of regression analysis and classification analysis [60,61,62,63] is the comparison of future and current values. The difference is that the regression analysis emphasizes the future value forecast while the classification research focuses on the future direction of rising and falling prices. The traditional data labeling methods emphasize the micro-level prediction of rising and falling prices. Thus, the corresponding data labeling method is used to label the data through the difference of data at different time points.

However, the short-term randomness of time series data is obvious, which is greatly influenced by various factors. Most existing time series forecasting models follow logical rules based on the relationship between adjacent states, and the inconsistency of fluctuations in the relevant time period is not considered [64]. Zhao et al. applied their method to five-minute high-frequency data and the daily data of Chinese stock markets, and the results showed that the logarithmic change of stock price (logarithmic return) has a lower possibility of being predicted than the volatility [65]. It is inferred that a model trained based on the data labeled by the comparison of the logarithmic rate of return will not have good prediction results. Since the continuous trend of time series data is often affected by intrinsic factors, it is more sustainable and predictable. For example, the rising or falling stock prices are often influenced by the company’s intrinsic value and macroeconomic environment, and it is difficult to forecast the rising and falling of stock prices on a certain day, while the forecast of the continuous trend is relatively reliable. The rising and falling of commodity prices are often influenced by macroeconomic factors and demand, and the forecast of trends is more feasible. The foreign exchange market is influenced by interest rates and international environment, and its trend often shows continuity in time. The intrinsic factors lead to the persistence and predictability of the trend, and the possibility of trend changing due to random factors is relatively low in short-term.

Therefore, this paper proposed a novel method to define the extracted features of the continuous trends of time series, aiming to make corresponding decisions by predicting the trend changes of time series. The prediction of continuous trends is more in line with the rules of financial time series data operation, more concerned with the prediction of trend change, and more in consistent with the investment habits of individual investors. In addition, an algorithm for automatically labeling data according to a given parameter was presented to automatically label the data used to train machine learning models. The prediction results were compared with those obtained by the traditional time series data labeling method. The classification result of the automatically labeling algorithm was significantly better than that of the traditional labeling method. The investment strategies were constructed, and the investment results were compared. The method proposed in this paper achieved much better performance.

The paper is organized as follows: In Section 1, the labeling method proposed in this paper was explained in details along with description of the automatic labeling algorithm. In Section 2, different values of parameter ω were used for data labeling and the results of different parameters in four groups were analyzed. In Section 3, the classification results of six machine learning models under different labeling method were compared and investment strategies were constructed based on the proposed labeling method to compare the results obtained from traditional labeling method and the buy-and-hold strategy. Finally, a short discussion is provided in Section 4 and the conclusions in Section 5. A detailed flowchart of the study is shown in Figure 1.

## 2. Methodology

### 2.1. Learning Algorithms

In order to better verify the effectiveness of the proposed labeling algorithm in this paper, six different machine learning models were selected to design experiments, including four traditional machine learning models and two deep learning models that are suitable for processing sequence data. These machine learning models can be briefly described as follows:

#### 2.1.1. Logistic Regression (LOGREG)

Logistic regression is a mathematical modeling approach that can be used to describe the relationship of several variables to a dichotomous dependent variable [66,67]. LOGREG theory is simple and easy to understand, but it lacks robustness and accuracy when there is noise in the data [68].

#### 2.1.2. Random Forest (RF)

The random forest classifier is an integrated classifier which generates multiple decision trees using randomly selected subsets of training samples and variables [69]. The random forest algorithm proposed, by Breiman in 2001, has been extremely successful as a general-purpose classification and regression method [70]. RF is one of the most powerful ensemble methods with high performance when dealing with high dimensional data [71].

#### 2.1.3. KNN

The K-nearest-neighbor (KNN) is a non-parametric classification method, which is simple but effective in many cases [72,73,74]. It can identify many market conditions including both mean reversion and trend following [75].

#### 2.1.4. Support Vector Machine (SVM)

Support Vector Machine (SVM) is a very special learning algorithm, which is characterized by the capacity control of decision function, the use of kernel functions and the scarcity of the solution [76]. It has many advanced features with good generalization capabilities and fast computing capabilities [77]. The support vector machine adopts a risk function composed of empirical error and a regularization term derived based on the principle of structural risk minimization, which is a promising method for predicting financial time series [78]. The fundamental motivation for SVM is that the method can accurately predict time series data when the underlying system process is usually a non-linear, non-stationary, and undefined priori [79]. The SVM kernel mechanism can map nonlinear data into high latitude space and make it linearly separable. SVM has a good effect on time series classification prediction [80].

#### 2.1.5. Long Short-Term Memory (LSTM) 

LSTM is a kind of time recurrent neural network, which is suitable for processing and predicting important events with a relatively long interval and delay in time series. It was first proposed by Hochreiter and Schmidhuber in 1997 [81]. The ingenuity of LSTM is that the weights of the self-loop could be changed by increasing the inputting gate threshold, forgetting gate threshold and outputting gate threshold. In this way, when the model parameters are fixed, the integral scale at different time points can be dynamically changed, thereby avoiding gradients vanishment [82,83,84].

#### 2.1.6. Gated Recurrent Unit (GRU)

GRU is a variant of LSTM, which was first proposed by Cho et al. in 2014 [85]. The difference between GRU and LSTM is that one gate threshold is used to replace the inputting gate threshold and the forgetting gate threshold, that is, an “updating” gate threshold is used to control the state of the cell. The advantage of this method is that the calculation is simplified and the expression ability of the model is excellent (Table 1) [86,87]. The parameters of the models above are shown in Table 1.

### 2.2. Vector Space Reconstruction

#### 2.2.1. Vector Dimension Expansion

Time series data gradually accumulate with the passage of time, and a sliding window parameter *λ* is needed. In order to allow the data vector to keep certain historical information, the vector dimension is extended according to the sliding window parameter. The purpose of dimension expansion is to enable the current vector to contain the historical price information in the length of the sliding window. The sliding window parameter *λ* was set to 11 in this paper and was used as an empirical parameter in combination with the trend characteristics of China’s stock market. In this paper, the closing price was selected for feature processing and model training.

The vector dimension expansion was carried out after determining the parameter of the sliding window. The formulas were shown in Equations (1) and (2), where *x*, *X*, *y*, *x_i_* and *y_i_* represent the raw data, the expanded matrix data [88], the label vector of the expanded data *X*, the closing price on the ith day, and the *i*-th label of the extended vectors, respectively. After dimension expansion, the one-dimensional data with only one closing price on that day were extended to *λ*-dimensional data:(1)$$x=\;\left[\begin{array}{l} x_{1}\;\\ x_{2}\\ .\;\\ .\;\\ x_{N-1}\;\\ x_{N}\; \end{array}\right]\to X=\;\left[\begin{array}{lllll} x_{\lambda } & x_{\lambda -1} & x_{\lambda -2} & \ldots & x_{1}\\ x_{\lambda +1} & x_{\lambda } & x_{\lambda -1} & \ldots & x_{2}\\ . & . & . & \ldots & .\\ . & . & . & \ldots & .\\ x_{N-1} & x_{N-2} & x_{N-3} & \ldots & x_{N-\lambda }\\ x_{N} & x_{N-1} & x_{N-1} & \ldots & x_{N-\lambda +1} \end{array}\right]$$
(2)$$y=\left[\begin{array}{l} {label}_{\lambda }\\ {label}_{\lambda +1}\\ .\;\\ .\;\\ {label}_{N-1}\\ {label}_{N} \end{array}\right]$$

#### 2.2.2. Feature Processing Method without Look-Ahead Bias

A key step in data preprocessing is data standardization or normalization. Traditional time series normalization or standardization processes often need to access all data, and there is usually a look-ahead bias [57,89]. This paper proposed a new feature extraction method, which retained the relevant information while reducing the absolute data size. At the same time, the original data could be scaled to a relative stable range of fluctuation. The feature vector based on the mean-deviation-rate was associated with historical data only and did not have look-ahead bias. Instead of standardization or normalization based on all data, the sliding window parameter *λ* was used to dynamically calculate a mean-deviation-rate by deviating from the mean value, and solving the problem of look-ahead bias. The formula was as follows: For the *λ*-dimensional vector, since the *λ* –1 length of data was the historical closing price, the mean value of this vector was subtracted and the value of each data was divided by this mean value. The feature $f_{ij}$ was computed as Equations (3) and (4):(3)$$f_{ij}=(x_{ij}-{M^{\lambda }}_{i})/{M^{\lambda }}_{i},x_{ij\;\in \;}X$$
(4)$${M^{\lambda }}_{s}=\frac{\sum_{i=s}^{s+\lambda -1}x_{i}}{\lambda },x_{i}\in x,s=1,2\ldots N-\lambda +1$$
where *x_ij_* denotes the closing price in the data matrix *X* and *M^λ^_s_* denotes the mean of the closing price of numbers in the corresponding sliding window period *λ*. After preprocessing of the original data by the above formula, the feature matrix *F* was obtained (the dates of the data were not listed) as follows Equation (5):(5)$$F=\;\left[\begin{array}{llll} f_{1,1} & f_{1,2} & \ldots & f_{1,\lambda }\\ . & . & \ldots & \\ . & . & \ldots & \\ f_{N-\lambda ,1} & f_{N-\lambda ,2} & \ldots & f_{N-\lambda ,\lambda }\\ f_{N-\lambda +1,1} & f_{N-\lambda +1,2} & \ldots & f_{N-\lambda +1,\lambda } \end{array}\right]$$

### 2.3. Definition of Continuous Trend Labeling

When the market evolves with a continuous trend, it is divided into a rising market and a falling market. The investors should buy and hold the target (stocks or commodities) in the rising market but hold the short position in a market with short mechanism. If there is no short mechanism, investors should sell the target in the falling market. Their position should not change until the forecast of the market trend is about to change [90]. In order to distinguish the continuous trends, this paper provided the following definition of continuous rising and continuous falling trends. First, the peak points and the trough points of the historical data in a time period were put into vectors *h* and *l*, where *t* represents the number of peak points, and *m* represents the number of trough points in Equation (6). A TD index was used to calculate the degree of trending of some time series data, and the result of the index reflected the continuous trend fluctuation degree of two adjacent peak and trough points in Equations (7) and (8):(6)$$h=\;\left[\begin{array}{l} h_{1}\\ h_{2}\\ .\\ .\\ h_{t-1}\\ h_{t} \end{array}\right]\ldots l=\left[\begin{array}{l} l_{1}\\ l_{2}\\ .\\ .\\ l_{m-1}\\ l_{m} \end{array}\right]$$
(7)$$TD(h_{i}l_{i-1})=abs(\frac{h_{i}-l_{i-1}}{l_{i-1}}),i>1$$
(8)$$TD(l_{i}h_{i-1})=abs(\frac{l_{i}-h_{i-1}-1}{h_{i-1}}),i>1$$

In this paper, by comparing the fluctuation parameter *ω* with the index value of TD, the continuous trend was defined as the fluctuation amplitude between the two adjacent peak and trough points exceeding the given threshold parameter *ω*, otherwise the fluctuation was considered a normal fluctuation without a continuous trend. The latest lowest and highest prices were selected as the basis of calculation, and the rise of the market above or the fall of the market below the *ω* parameter was defined as a continuous rising trend or a continuous falling trend, respectively. Then, all data labels were set to 1 in a period of the upward trend and −1 (or 0 for the training process of deep learning models, if the label is negative, it may report an error) in a period of the downward trend for model training. An example is given in Figure 2, and the time series data are a part of the data to be analyzed in the next step of the paper. 

The calculation of TD of L4H5 and H8L9 is shown in the figure. From the perspective of actual performance, it was hoped that L4H5(H8L9) would be viewed as an overall continuous upward(downward) trend, and the corresponding data label belonged to a same category, which was more in line with the law of market operation. However, the traditional labeling method in the L4H5(H8L9) section was obviously noisy and did not conform to the law of market operation, and the same was true for the rest of the series.

The trend of a market is often the combined results of fundamentals and economic environment. The duration of a trend is relatively long, it is more in line with the actual investment behavior in terms of actual operation. The traditional research methods focus on predicting the rising and falling prices in a short time period in the future, by evaluating the regression or classification effect of an established model, thus only focusing on the prediction of direction of short-term fluctuations. In practice, the traditional research methods are often unworkable, especially when the actual operation cost and market capacity are taken into account. Therefore, it is theoretically feasible to define the continuous up and down trends of the market and adopt a machine learning model to predict the direction of the market trend while ignoring the normal fluctuations, which is more in line with the law of financial market operation.

The label vector set *y* is obtained from the labeling operations carried out after the parameter *ω* is given. However, different investors have different judgments on the continuous trend even for the same stock or commodity market, so the way of labeling historical data is also unique. The reason is that different capitals, risk tolerances, investment decision-making cycles and other factors lead to diverse investment methods and investment styles. Thus, different investors have distinct definitions of the continuous trend of the market. As a result, the investors can label historical data with unique parameter and train models to get the most suitable model to guide their investment. In this paper, an automatic labeling algorithm was proposed.
**Algorithm 1. Auto-Labeling Data for CTL****Input**: Original Time series data$X={\left[x_{1},x_{2},x_{3},...,x_{N}\right]}^{T}$,ω>0, which represents the proportion threshold parameter of the trend definition**Output**: The label vector $Y={\left[{label}_{1},{label}_{2},{label}_{3},...,{label}_{N}\right]}^{T}$**Initialization of related variables:***FP*=*x_1_*, which represents the first price obtained by the algorithm; *x_H_ =x_1_*, used to mark the highest price; *HT=t_1_*, used to mark the time when the highest price occurs; *x_L_ =x_1_*, used to mark the lowest price; *LT=t_1_*, used to mark the time when the lowest price occurs; *Cid=0*, used to mark the current direction of labeling; *FP_N =0*, the index of the highest or lowest point obtained initially.***for****i = 1:N*
  ***if** (x_i_ > FP + x*_1_*ω*)*
    *Set [x_H_, HT, FP_N, Cid ] = [ x_i_, t_i_, i,1] and end for*

   ***if** (x_i_ < FP- x*_1_*ω*)*    *Set [x_L_, LT, FP_N, Cid ] = [ x_i_, t_i_, i, −1] and end for*
***end for i******for** i = FP_N+1:N*
   ***if** (Cid > 0)*
    ***if** (x_i_ > x_H_)*
     *Set [x_H_, HT] = [ x_i_, t_i_ ]*    ***if** (x_i_ < x_H_ - x_H_*ω and LT <= HT)*
     ***for** j = 1:N*
      ***if** (t_j_ > LT and t_j_ <= HT)*
       *Set y_j_ = 1*
     ***end for j***
      *Set [x_L_, LT, Cid] = [ x_i_, t_i_, −1]*
    ***if** (Cid < 0)*
     ***if** (x_i_ < x_L_)*
      *Set [x_L_, LT] = [ x_i_, t_i_ ]*     ***if** (x_i_ > x_L_ + x_L_ *ω and HT <= LT)*
     ***for** j = 1:N*
      ***if** (t_j_ > HT and t_j_ < = LT)*
       *Set y_j_ = −1*
     ***end for j***

     *Set[x_H_, HT, Cid] = [ x_i_, t_i_, 1]****end for i***

When the market rises above a certain proportion parameter *ω* from the current lowest point or recedes from the current highest point to a certain proportion parameter *ω*, the two segments are labeled as rising and falling segments, respectively. The labeling result is unique as long as the proportion threshold parameter *ω* is given for the labeling process. The value of *ω* in this paper was 0.15, which was obtained from the following analysis in the paper. The algorithm for the automatic labeling process based on a given parameter *ω* is presented in Algorithm 1.

## 3. Research Design

### 3.1. Data Description

This paper mainly analyzed the trend changes in China’s Shanghai Stock Composite Index (Stock Code 000,001) and Shenzhen Component Index (Stock Code 399,001) and forecasted the trend of the stock market to guide the investment analysis.

Shanghai Composite Stock Exchange Index (SSCI) is short for “Shanghai Stock Index” or “Shanghai Stock Composite Index”. Its component stocks are all listed on the Shanghai Stock Exchange, including A and B stocks. SSCI reflects the changes of the stock prices on the Shanghai Stock Exchange. SSCI was officially released with a base value of 100 on 19 December 1990 and has since reflected the overall trend of the Chinese stock market [91].

Shenzhen Component Stock Exchange Index (SZCI) refers to the weighted composite stock price index compiled by the Shenzhen Stock Exchange and covers all stocks listed on the Shenzhen Stock Exchange. The Shenzhen Component Index was compiled and published by the Shenzhen Stock Exchange on 3 April 1991 with a base index of 100 points.

As the barometer of China’s economy, Shanghai Stock Index and Shenzhen Component Index have a profound impact on all walks of life. China’s stock market provides financing facilities for enterprises and is crucial for the development of these enterprises. SSCI and SZCI play an important role in the national economic development at macro and micro levels and have great significance in the prediction of important index trends. Therefore, the correct prediction of the trend of China’s stock market not only can guide the formulation of relevant economic policies, but also can better prevent financial risks, facilitate the dynamic flow of capital, and allow funds to flow to enterprises in need.

This paper mainly analyzed the trend of two stock indexes in China by using daily stock transaction data collected from China RichDataCenter (the data can be downloaded from GitHub (https://github.com/justbeat99/Daily-Stock-Data-Set) or China RichDataCenter (http://www.licai668.cn/content/showContent.asp?titleNo=365)) based on the backward answer authority, which is the price before eliminating advantageous position carries when the answer authority changeless, raise the price after eliminating advantageous position. In order to further compare the method proposed in this paper, three stocks with sufficient data were also randomly selected to test the trend prediction ability of the models. Then, investment strategies were established to test the investment performance. These three stocks were those of the Founder Technology Group Corp. (stock code 600,601), Shenzhen Cau Technology Co., Ltd. (stock code 000,004), and Shanghai Fenghwa Group Co., Ltd. (stock code 600,615).

### 3.2. Input Setup

Combined with the trading rules and the volatility level of China’s stock market, the sliding window parameter *λ* = 11 was obtained as an empirical parameter. In order to be comparable, all parameters of the machine learning models remained the same for the comparison experiments and were not optimized. In the segmentation of training and test sets, it was considered to establish strategies for the later steps to obtain the profit gain/loss results for the net yield rate curve, and the larger the amount of test data, the more convincing the results. At the same time, the larger the training set, the better learning performance of the models. Thus, this paper adopted manual segmentation of training and test sets to balance the above problems by splitting the test and training sets in a ratio close to 1:1 (In fact, it has been verified that the models have converged after the training set data are close to 1000). The actual data available for the training set would be subtracted by *λ* − 1 raw data owing to the sliding window parameter *λ*. The data in the first length of *λ* − 1 contained no more historical data for expansion, so they could not be used for training. The relevant data are shown in Table 2.

### 3.3. Comparison Experiments

In order to compare the prediction results of the labeling method proposed by this paper with those obtained from the traditional time series data labeling method, four groups of comparative experiments were designed. Since this paper constructed investment strategies based on the forecasting direction, the regression analysis was also reduced into high-low direction forecasting problems to compare with the method proposed by this paper. The traditional labeling method labeled data by comparing the closing price *X_t_* + *m* at a certain time with the closing price *X_t_*. In the comparative experiments of this paper, *m* = 1, 3, 5, 10. Table 3 showed the labeling process of the labeling method proposed by this paper and the comparison experiments, where *X_t_* represents the closing price at time *t*, “Label_t_” represents the Label of the *t*-th data. *E* was the experiment of the labeling method proposed by the paper, and C1, C3, C5, C10 represent the comparative experiments, respectively.

### 3.4. Statistical Metrics

In order to evaluate the prediction efficiency of the trained models, six statistical metrics, namely accuracy, recall, precision, F1_score, AUC and NYR were selected to evaluate the prediction classification results [92,93]. The former five metrics are classification metrics that would be maximized when the model did not generate false positive or false negative predictions, as shown in the Table 4 below [94], NYR is the profit metric.

True positives (TP) denote the success in the identification of the correct reinforcement class (positive samples), true negatives (TN) denote the successful classification of negative samples, false positives (FP) stand for the incorrect classifications of negative samples into positive samples, and false negatives (FN) denote the positive samples that are incorrectly predicted as negative samples [92,95]. Acc is the most primitive evaluation metric in classification problems. The definition of accuracy is the percentage of the correct results in the total sample. In terms of AUC, $x_{i}^{+}$ and $x_{i}^{-}$ represent the positive and negative labels of data points, respectively. *f* is a generic classification model, 1 is an indicator function equals to 1 when $f\left(x_{i}^{+}\right)\geq f\left(x_{j}^{-}\right)$ and 0 otherwise, *N*^+^ (resp., *N^−^*) is the number of data points with positive (resp., negative) labels, and *M = N*^+^*N*^−^ is the number of matching points with opposite labels (*x*^+^,*x*^−^), with a value ranging from 0 to 1. The higher the value, the better the model. A random guess model has 0.5 as its AUC value [96]. NYR represents the cumulative return on the investment strategy, where *R_j_* denotes the daily return of a stock, *HD* stands for the number of days when the positions of the stock are held in a buy-and-sell process, and *NT* represents the total number of transactions of a buy-and-sell strategy.

## 4. Experimental Results

### 4.1. Analysis of Threshold Parameters

The selection of the parameter *ω* was based on the different situations of actual investors (including but not limited to investment capital, risk tolerance, trading frequency, etc.) and the fluctuation of the corresponding market target. In order to objectively evaluate the proposed labeling method, a more objective method was used to determine the value of parameter *ω*, and the traditional four machine learning models were used for the analysis and study of parameter *ω*. In this study, different *ω* parameters were compared and analyzed, and a parameter *ω* with relatively better classification results was finally chosen as the basis of the next comparative experiment and strategy construction. The value of parameter *ω* was set to 0.05–0.5 with a step size of 0.05 to label SSCI and SZCI data based on the automatic labeling algorithm proposed in this paper, and four traditional machine learning models were trained accordingly. Figure 3 shows the specific graphic results of the classification metrics of the four models assigned with different values of parameter *ω* based on the SSCI and SZCI data. In the training process, the 10-fold cross-validation was used to train the model, and a mean accuracy value of 10-fold cross-validation was obtained.

It can be seen from plots a to e in Figure 3 that while the parameter *ω* gradually increased from 0.05 to 0.5, the values of accuracy, precision and AUC of the classification results gradually decreased, while the values of recall and F1_score did not decrease significantly. When the parameter *ω* was between 0.05 and 0.25, the values of accuracy and AUC remained above 0.6 and the classification accuracy was relatively high, indicating that the ROC curve worked well and the precision was above 0.55. The results obtained from plot f were consistent with the results obtained from plots a to plot e, indicating that the classification performance was the best when the parameters were between 0.05 and 0.25. Therefore, the parameter *ω* should be close to 0.05. However, in order not to overemphasize the role of parameter *ω*, a suboptimal parameter value *ω* = 0.15 was chosen instead and the average value ranged from 0.05 to 0.25. Therefore, if the price rose or fell by more than 15%, it would be regarded as an upward or downward trend, respectively, and the data would be labeled by the automatic labeling algorithm accordingly.

### 4.2. Classification Results and Analysis

In this part, the trained LOGREG, RF, KNN, SVM, LSTM, GRU models with the parameter *ω* = 0.15 were used to compare the classification results of the two stock indexes and three stocks. In order to distinguish the effects of the traditional four machine learning models and two deep learning models, the experimental results were listed separately. Table 5 showed the results of average accuracy of the 10-fold cross-validation of the four traditional machine learning models carried out using the training set. The “Average_Accuracy” column represented the average accuracy of the four traditional machine learning models using the same experimental results of the same stock. The value of “Average_Accuracy” could more objectively reflect the experimental results of the various labeling methods. It can be clearly seen that the average classification accuracy of experiment E on all stock indices and stocks was close to 0.7 of the four machine learning models, much exceeding the results of C1, C3, C5, and C10, and was consistent with that of “Average_Accuracy”.

The average accuracy of experiments C1, C3, C5 and C10 was significantly lower. Except for the average accuracy value of C1 experiment of 600,601 that was above 0.6, all other experimental results were between 0.51 and 0.56, indicating much poorer performance than the results of the proposed method. Thus, the labeling method proposed by this paper was more in line with the characteristics of the market and more valuable for the models. The four traditional machine learning models trained by the training set generated by the labeling method proposed in this paper showed obvious better results.

The effects of the trained models on the test set were also verified. Table 6 shows the corresponding AUC values. It can be clearly seen that the AUC values of experiment E exceed the results of C1, C3, C5, and C10 in all stocks tested a lot, indicating that the data trained by the automatic labeling algorithm can be well generalized with excellent learning performance. Moreover, the results of C1, C3, C5, and C10 were basically around 0.5, although the result of C1 of 600,601 was above 0.6, indicating poorer performance than that of experiment E.

All ROC curves were drawn to provide better visual effects. However, since the space was limited, only the AUC values were provided here in Table 6.

The four metrics of the classification results, precision, recall, accuracy and F1_score of the models KNN, LOGREG, RF, and SVM, are given in Table 7 and Table 8, respectively. Meanwhile, the average of the results of the four model-related metrics used to evaluate the test results more objectively was obtained from Table 9. In Table 7, it can be see that all the index values corresponding to test E as well as the values of Accuracy and F1_score were the highest among all model results obtained from the test set of two stock indices SSCI and SZCI and three stocks, and the classification results of accuracy were higher than those of C1, C3, C5, and C10. As far as the value of metric Precision was concerned, the value of experiment E corresponding to 600,601 of the model LOGREG was 0.6881, which was slightly lower than the corresponding C1 result of 0.6899, but the difference was not significant.

The precision value of experiment E of 000,004 in model LOGREG was 0.7889, which was slightly lower than the corresponding C1 value of 0.8571. The precision value of experiment E of 600,601 in the model SVM was 0.6942, which was slightly lower than the corresponding value 0.7055 of C1. 

As far as the value of recall was concerned, the value of experiment E in models LOGREG and SVM was slightly lower than that in the comparison experiment, but the other values were the highest among all recall values. Moreover, the data showed that the corresponding F1_score of C1, C3, C5 and C10 of 000,004 and 600,615 were very low for LOGREG and SVM, indicating that the machine learning models have not learned effective patterns using the training data in experiments C1, C3, C5 and C10, and showed poor performance in the test set.

In order to better analyze the results, the differences of machine learning algorithms, as well as the situations caused by the sample differences, were taken into account. Then, the corresponding classification metrics of the four machine learning models were averaged, as shown in Table 9. The table clearly showed that all corresponding index values of experiment E were the highest, and the average Accuracy values were basically around 0.6–0.7, indicating the model had a high accuracy in predicting market trends of the test set. The experimental results showed that the labeling method proposed in this paper was in line with the law of the market, suitable for labeling time series data with trend characteristics. In addition, the machine learning models trained in this study showed good performance in prediction.

Table 10 shows the classification results of the two deep learning models of LSTM and GRU. It can be clearly seen that the accuracy values of the two deep learning models in SSCI and SZCI exceed 0.7, which are basically around 0.72. The accuracy results are better than the traditional four machine learning models, and the other four metrics are also significantly better than the traditional four machine learning models. The accuracy results of 600,615 even reach 0.74 and 0.75 on LSTM and GRU, respectively. Concerning LSTM and GRU, the difference of the metrics is not significant. The overall results are better than those of the four traditional machine learning algorithms. Moreover, the results of all metrics values of E much surpass the results of C1, C3, C5, and C10, confirming the effectiveness of our proposed algorithm again.

### 4.3. Implementation of Strategies

In this part, the trend prediction results of the models using the test set were constructed as investment strategies to test the investment efficiency of the models based on different labeling methods. Considering the price changes caused by market fluctuations and in order to reflect the investment process as close as the actual situation, in conjunction with the available amount of funds and purchasable volume, the issue of positions should also be taken into account. The reason was that there would be a problem when the account balance was lower than the initial balance, so that the positions available for purchase would be reduced. Based on the above considerations, this paper proposed the following hypotheses to construct the strategies:

**Hypothesis** **1.**
*SSCI and SZCI were tradable stock indexes. The profit and loss were settled according to the absolute index point, and the contract multiplier was 1.*


**Hypothesis** **2.**
*The strategies constructed by experiments E, C1, C3, C5, and 10 and the buy-and-hold strategy would be compared with each other based on the net yield rate.*


The initial balance for the strategies of the models and the buy-and-hold investment strategy was one million. The investment strategies constructed by the models would have a full load in stock index contracts or stocks in the upward trend and would sell all contracts or stocks in the downward trend, and the actual amounts that could be bought or sold would be calculated according to the balance. The buy-and-hold investment strategy bought the stock index contracts or stocks at the very beginning and held them until the end of the test period, when the stock index contracts or stocks would be sold.

For the strategy construction, based on the two hypotheses above, the investment strategies constructed by the models would conduct buying and selling operations based on the predicted labels of the stock data. In fact, the traditional regression algorithm ultimately built the strategies according to the direction of forecasted price fluctuations, thus the strategies based on C1, C3, C5, and C10 could be considered to cover the strategies of the regression prediction construction. Since there is no short-selling mechanism in China’s stock market (without considering the security lending mechanism for the time being), the strategies constructed by the models would buy stock index contracts or stocks in the predicted rising trend and sell all holdings when the market was about to decline. The strategies were constructed as follows: if the label of classification prediction was 1, the buying operation would be performed; if the label of classification prediction was - 1, the selling operation would be performed.

Table 11 shows the investment results of two stock indices SSCI and SZCI and three stocks of the strategies of the four traditional machine learning models, including the buy-and-hold strategy. In terms of the results of Average_NYR, the net yield rate of experiment E of all stock indices and stocks was the highest, much exceeding that of the buy-and-hold strategy. The Average_NYR values of C1, C3, C5, and C10 of SSCI and SZCI exceeded the results of the buy-and-hold strategy but were lower than the results of Experiment E. In terms of the Average_NYR values of C1, C3, C5, and C10 of 600,601 and 000,004, some values were larger than those of the buy-and-hold strategy and some values were lower than those of the buy-and-hold strategy, without obvious robustness and stability compared to the results of experiment E. Specifically, the Average_NYR result of 600,615 was the only result of experiment E, exceeding the result of the buy-and-hold strategy, and the values of C1, C3, C5, and C10 were all far below that of the buy-and-hold strategy. Specifically, for each of the four machine learning models, it can be clearly seen that the results of experiment E were the best, especially for the results of 600,601 and 600,615, both of which much exceeded those of C1, C3, C5, and C10. It can also be seen from the data that the net yield rate to maturity of 000,004 and 600,615 in experiment E was much higher than the corresponding results of C1, C3, C5 and C10. The results from the net yield rate to maturity proved the optimality of the proposed labeling method.

Table 12 shows the investment results of two stock indices SSCI and SZCI and three stocks of the strategies of LSTM and GRU. Concerning SSCI and SZCI, the value of Average_NYR of E not only much exceeds the results of C1, C3, C5, and C10, but also exceeds the traditional four machine learning models. In terms of the individual stock profit rate, only the stock code 600,601 has a yield rate of 412.25%, which exceeds the average result of the traditional four machine learning models. However, considering that the SSCI and SZCI are more representative, the differences of individual stocks are relatively large and the results of the previous classification metrics, it is believed that LSTM and GRU are better than traditional machine learning models in processing time series data classification. The results once again confirm the superiority of our labeling method.

The net yield rate curves for SSCI and SZCI of the four traditional machine models and the two deep learning models are shown in Figure 4 and Figure 5, respectively. The curves of experiment E were almost above all other curves at each time point, indicating that the profitability of experiment E was higher than that of other experiments. In fact, only the net yield rate result of C5 in subfigure d was similar to the corresponding result in experiment E in Figure 4. In terms of the net yield rate to maturity, it can be seen that the corresponding results of experiment E were still the best. The results of subfigure h in Figure 4 showed that the performance of SVM in C1, C3, C5 and C10 was poor not only due to the difference in variety, but also due to the differences in algorithm models and parameters. At the same time, it was noticed that when the stock indexes SSCI and SZCI rose as a whole, each strategy made different levels of profit. While SSCI and SZCI fell as a whole, each strategy showed a reduced net yield rate. Therefore, if the model could better predict the trend, it could buy the corresponding target when the market trend was about to rise but sell the target in advance when the market trend was about to fall.

The experimental results showed that the models based on the labeling method proposed in this paper achieved better results than the traditional labeling method for time series trend prediction. The results demonstrated that the labeling method proposed in this paper was superior to the traditional labeling method for locating the trend characteristics of financial time series data.

## 5. Conclusions

This paper proposed a novel data labeling method called CTL to extract the continuous trend feature of financial time series data. In the feature preprocessing stage, this paper proposed a new method that can avoid the problem of look-ahead bias encountered in the traditional data standardization or normalization process. Then, an automatic labeling algorithm was developed to extract the continuous trend features of financial time series data, and the extracted trend features were used in four supervised machine learning methods and two deep learning models for financial time series prediction. The experiments performed on two stock indexes and three stocks demonstrated that CTL was superior to the state-of-the-art data labeling method in terms of classification accuracy and some other metrics. Furthermore, the net yield rate obtained by the strategies built on the financial time series prediction of CTL was much higher than that of other labeling methods, and far exceeded that of the buy-and-hold strategy, which represents the maturity return of the index itself.

## Figures and Tables

**Figure 1 entropy-22-01162-f001:**
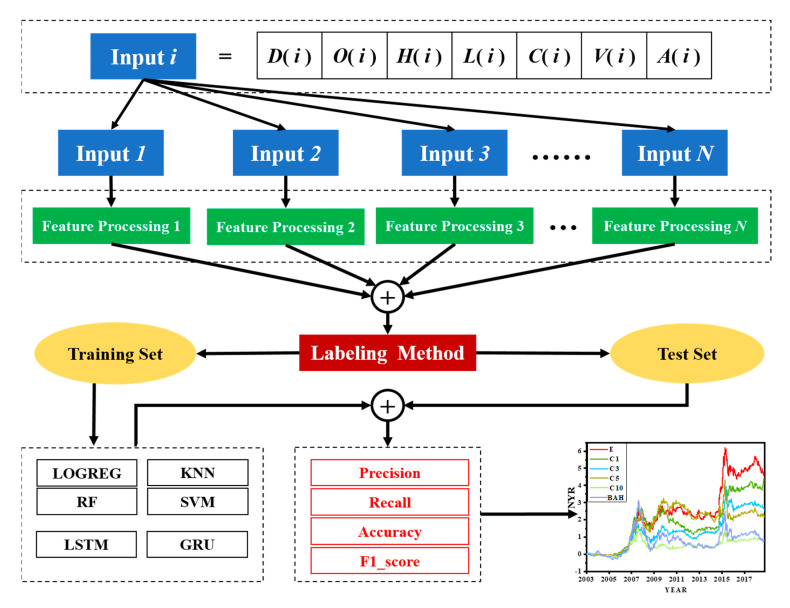
Flowchart of the steps of the proposed method. D(*i*), O(*i*), H(*i*), L(*i*), C(*i*), V(*i*), and A(*i*) represent the ith date, opening price, highest price, lowest price, closing price, volume, and quantity, respectively. The prediction results are obtained by the labeling method proposed in this paper, and the investment strategies are constructed.

**Figure 2 entropy-22-01162-f002:**
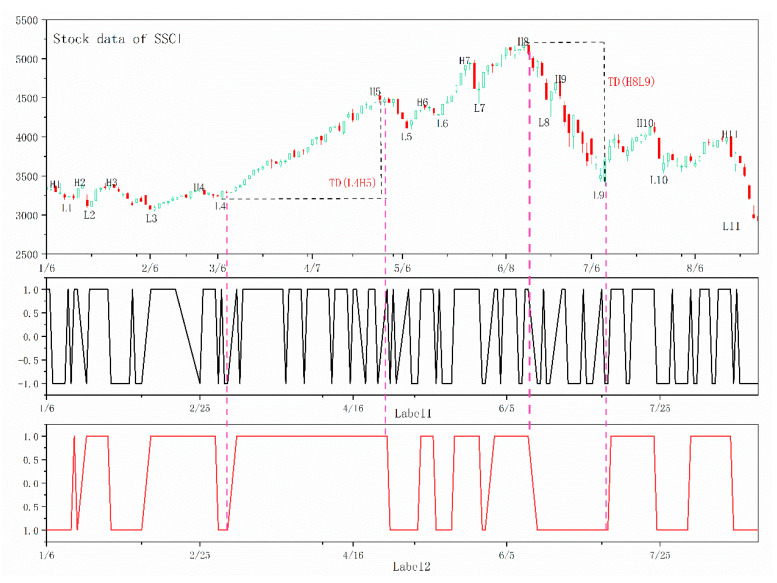
Definition of continuous trend labeling: the market was divided into two categories, rising market and falling market, based on the index of TD. The “label1” denotes the labeling results of traditional labeling methods. The “label2” represents the labeling results of labeling method based on the trend definition proposed in this paper. From the picture, we can see that in any period of time, such as the trend of L4H5, our labeling method gives one direction label for the data, but the traditional labeling methods label the data in two directions in this period of time, which is not in line with the reality. It is the same situation with H8L9.

**Figure 3 entropy-22-01162-f003:**
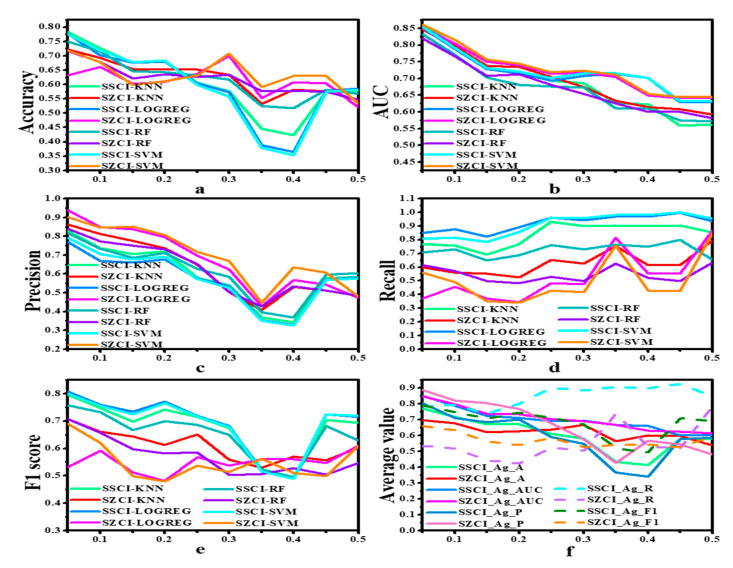
Classification results of four traditional machine learning models with different threshold parameters based on the data of SSCI and SZCI. The six figures represent the values of Acc, AUC, P, R, F1, Average_value under different thresholds respectively. Ag represented “average”, A represents accuracy, P represents precision, R represents recall, and F1 represents F1_score. The *X* axis represents the value of the parameter, and the *Y* axis represents the corresponding classification index value. In the results, to balance the results of the four traditional machine learning models, plot f averaged the classification results of the four machine learning models. It can be seen from the picture that a threshold parameter set at 0.1–0.2 is better.

**Figure 4 entropy-22-01162-f004:**
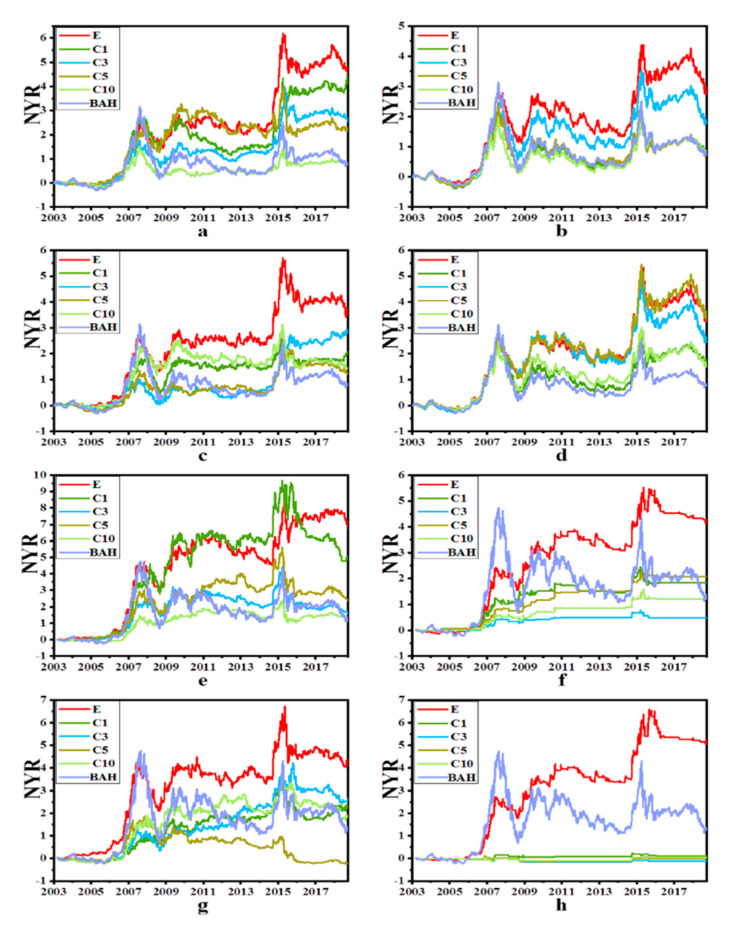
NYR Curves of SSCI and SZCI. (**a**–**d**) represent the results of KNN, LOGREG, RF, SVM of SSCI, respectively and (**e**–**h**) represent the results of KNN, LOGREG, RF, SVM of SZCI respectively. The X axis is date and the Y axis is the net yield rate. “BAH” is short for “buy-and-hold” strategy. It can be seen from the figure that the cumulative rate of return at each time point of different experiments on the index.

**Figure 5 entropy-22-01162-f005:**
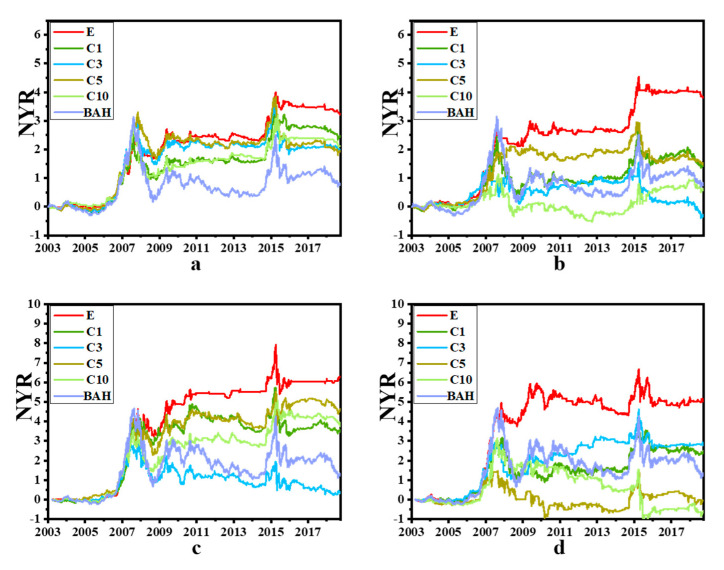
NYR Curves of SSCI and SZCI. (**a**,**b**) represent the results of LSTM and GRU of SSCI respectively, (**c**,**d**) represent the results of LSTM and GRU of SZCI respectively. The X axis is date and the Y axis is the net yield rate. “BAH” is short for “buy-and-hold” strategy. It can be seen from the figure that the cumulative rate of return at each time point of different experiments on the index.

**Table 1 entropy-22-01162-t001:** The related parameters information of the six models mentioned above in this paper.

Models	Related Parameters
LOGREG	The penalty was set as “L2”.
RF	The maximum number of iterations parameter N was set as 10 in this study.
KNN	The parameter of N was set as 20.
SVM	The RBF kernel function was used and the regularization parameter *C* was set as 2, and the kernel parameter *σ* was also set as 2.
LSTM	Hidden size = 50; the number of layers = 2. The optimization function is Adam with parameter learning rate = 0.02, betas = (0.9, 0.999).
GRU	Hidden size = 50; the number of layers = 2. The optimization function is Adam with parameter learning rate = 0.02, betas = (0.9, 0.999).

**Table 2 entropy-22-01162-t002:** The segmentation of training and test sets.

Stock Code	Training Set Total Data Points	Training Set Duration	Test Set Total Data Points	Test Set Duration
000,001 China	3010	19 December 1990 to 7 March 2003	3846	10 March 2003 to 27 December 2018
399,001 China	3010	03 April 1991 to 20 May 2003	3799	21 May 2003 to 27 December 2018
600,601 China	3010	19 December 1990 to 30 June 2003	3722	01 July 2003 to 27 December 2018
000,004 China	3010	02 January 1991 to 28 May 2003	3474	29 May 2003 to 27 December 2018
600,615 China	3210	10 September 1992 to 27 July 2006	2769	28 July 2006 to 27 December 2018

**Table 3 entropy-22-01162-t003:** Comparison strategies for different experiments.

Experiment Name	Label*_t_* = 1	Label*_t_* = −1
E	Labeling Algorithm	Labeling Algorithm
C1	*X_t_*_+1_ > *X_t_*	*X_t_*_+1_ < *X_t_*
C3	*X_t_*_+3_ > *X_t_*	*X_t_*_+3_ < *X_t_*
C5	*X_t_*_+5_ > *X_t_*	*X_t_*_+5_ < *X_t_*
C10	*X_t_*_+10_ > *X_t_*	*X_t_*_+10_ < *X_t_*

**Table 4 entropy-22-01162-t004:** Metrics for classification or profitability evaluations.

Metrics	Formula	Evaluation Focus
Accuracy (Acc)	$\frac{TP+TN}{TP+FN+FP+FN}$	Indicating the percentage of correct predictions in all samples.
Recall(R)	$\frac{TP}{TP+FN}$	Indicating the proportion of positive samples classified as positive samples.
Precision(P)	$\frac{TP}{TP+FP}$	Indicating the proportion of actually positive samples among those classified as positive.
F1_score(F1)	$2\times \frac{Precision\times Recall}{Precision+Recall}$	F1 is the weighted harmonic mean of Precision and Recall.
AUC	$\frac{\sum_{i=1}^{N+}\sum_{j=1}^{N-}1f(x_{i}^{+})\geq f(x_{j}^{-})}{M}$	AUC can objectively reflect the ability of comprehensively predicting positive samples and negative samples and eliminate the influence of sample skew on the results to a certain extent.
NYR	$\sum_{i=1}^{NT}\sum_{j=1}^{HD}Rj$	In order to compare the profitability of the constructed strategies, the net yield rate NYR was used to evaluate the strategies.

**Table 5 entropy-22-01162-t005:** Average accuracy of 10-fold cross-validation of different experiments carried out on different stock training sets of the four traditional machine learning models.

Stock Code	Experiment Name	KNN	LOGREG	RF	SVM	Average _Accuracy
000,001	E	0.6853	0.7206	0.6799	0.7243	0.7025
	C1	0.5180	0.5180	0.5106	0.5127	0.5148
	C3	0.5666	0.5319	0.5343	0.5382	0.5427
	C5	0.5599	0.5216	0.5532	0.5565	0.5478
	C10	0.5450	0.5453	0.4987	0.5739	0.5407
399,001	E	0.6920	0.7040	0.6690	0.7087	0.6934
	C1	0.5343	0.5506	0.5226	0.5250	0.5331
	C3	0.5476	0.5510	0.5416	0.5340	0.5436
	C5	0.5663	0.5713	0.5690	0.5543	0.5652
	C10	0.5623	0.5770	0.5543	0.5630	0.5641
600,601	E	0.6930	0.7050	0.6613	0.7043	0.6909
	C1	0.5937	0.6837	0.5987	0.6787	0.6387
	C3	0.5504	0.6023	0.5277	0.5690	0.5624
	C5	0.5318	0.5676	0.5450	0.5528	0.5493
	C10	0.5170	0.5293	0.4987	0.5367	0.5204
000,004	E	0.7090	0.7220	0.7007	0.7237	0.7138
	C1	0.5257	0.5457	0.5450	0.5400	0.5391
	C3	0.5373	0.5407	0.5160	0.5363	0.5326
	C5	0.5289	0.5573	0.5186	0.5453	0.5375
	C10	0.5277	0.5457	0.5147	0.5513	0.5348
600,615	E	0.6775	0.6865	0.6693	0.7143	0.6869
	C1	0.5122	0.5219	0.5044	0.5228	0.5153
	C3	0.5134	0.5325	0.4959	0.5278	0.5174
	C5	0.5040	0.5353	0.5190	0.5215	0.5200
	C10	0.4990	0.5313	0.5068	0.5303	0.5169

**Table 6 entropy-22-01162-t006:** The AUC values of the classification results from the test sets of all models.

Stock Code	Experiment Name	KNN	LOGREG	RF	SVM
000,001	E	0.7302	0.7286	0.7035	0.7318
	C1	0.5209	0.5388	0.5195	0.5408
	C3	0.5272	0.5465	0.5249	0.5499
	C5	0.5399	0.5500	0.5210	0.5521
	C10	0.5554	0.5654	0.5299	0.5727
399,001	E	0.7371	0.7505	0.7076	0.7569
	C1	0.5167	0.5216	0.5230	0.5261
	C3	0.5280	0.5395	0.5246	0.5474
	C5	0.5331	0.5502	0.5288	0.5571
	C10	0.5302	0.5565	0.5109	0.5542
600,601	E	0.7343	0.7380	0.7131	0.7514
	C1	0.6593	0.7330	0.6582	0.7446
	C3	0.5523	0.5791	0.5475	0.5923
	C5	0.5596	0.5748	0.5447	0.5827
	C10	0.5596	0.5618	0.5437	0.5808
000,004	E	0.7471	0.7494	0.7215	0.7578
	C1	0.4920	0.4905	0.5027	0.4969
	C3	0.4952	0.4952	0.4959	0.4967
	C5	0.5196	0.5029	0.4971	0.5039
	C10	0.5277	0.5280	0.5139	0.5184
600,615	E	0.7960	0.7762	0.7701	0.8093
	C1	0.4967	0.5077	0.4759	0.4972
	C3	0.4880	0.5168	0.4838	0.5446
	C5	0.5001	0.5315	0.4855	0.5304
	C10	0.5138	0.5102	0.5066	0.5550

**Table 7 entropy-22-01162-t007:** The metrics values of the classification results from the test sets of KNN and LOGREG.

Stock Code	Experiment Name	KNN				LOGREG			
		P	R	Acc	F1	P	R	Acc	F1
000,001	E	0.7050	0.6919	0.6752	0.6984	0.6617	0.8234	0.6752	0.7337
	C1	0.5542	0.4583	0.5172	0.5017	0.5306	0.9446	0.5273	0.6795
	C3	0.5402	0.4688	0.5187	0.5020	0.5209	0.8829	0.5192	0.6552
	C5	0.5604	0.4511	0.5244	0.4999	0.5288	0.9279	0.5265	0.6737
	C10	0.5764	0.4802	0.5351	0.5240	0.5481	0.8790	0.5494	0.6752
399,001	E	0.7742	0.5500	0.6544	0.6431	0.8367	0.3691	0.6020	0.5123
	C1	0.5331	0.3621	0.5078	0.4313	0.5762	0.1313	0.5025	0.2138
	C3	0.5539	0.3376	0.5170	0.4195	0.6145	0.0519	0.4930	0.0958
	C5	0.5663	0.3322	0.5162	0.4187	0.6579	0.0878	0.4975	0.1549
	C10	0.5742	0.2902	0.5033	0.3855	0.6520	0.0799	0.4830	0.1424
600,601	E	0.6661	0.6642	0.6811	0.6652	0.6881	0.6501	0.6926	0.6686
	C1	0.6261	0.4805	0.6059	0.5437	0.6899	0.6372	0.6827	0.6625
	C3	0.5381	0.4201	0.5368	0.4718	0.5826	0.4943	0.5766	0.5348
	C5	0.5517	0.4973	0.5508	0.5231	0.5186	0.7120	0.5298	0.6001
	C10	0.5326	0.4865	0.5425	0.5085	0.4930	0.7598	0.5030	0.5980
000,004	E	0.7594	0.5802	0.6664	0.6578	0.7889	0.5411	0.6664	0.6420
	C1	0.4939	0.2310	0.4899	0.3148	0.8571	0.0034	0.4942	0.0068
	C3	0.4854	0.2094	0.4891	0.2925	0.3766	0.0165	0.4899	0.0317
	C5	0.5320	0.2599	0.5012	0.3492	0.4772	0.1230	0.4790	0.1956
	C10	0.5339	0.3635	0.5060	0.4325	0.5426	0.1451	0.4940	0.2289
600,615	E	0.8404	0.6831	0.7129	0.7536	0.8289	0.6180	0.6724	0.7081
	C1	0.5323	0.3219	0.4944	0.4012	0.5714	0.0027	0.4742	0.0055
	C3	0.5145	0.3099	0.4904	0.3868	0.5789	0.0077	0.4825	0.0151
	C5	0.5327	0.2985	0.4872	0.3826	0.6667	0.0068	0.4695	0.0134
	C10	0.5678	0.3302	0.4984	0.4176	0.5200	0.0172	0.4561	0.0334

**Table 8 entropy-22-01162-t008:** The metrics values of the classification results from the test sets of RF and SVM.

Stock Code	Experiment Name	RF				SVM			
		P	R	Acc	F1	P	R	Acc	F1
000,001	E	0.6856	0.6469	0.6469	0.6657	0.6759	0.7833	0.6781	0.7256
	C1	0.5482	0.4152	0.5083	0.4725	0.5339	0.9431	0.5330	0.6818
	C3	0.5377	0.4196	0.5130	0.4714	0.5269	0.8015	0.5250	0.6358
	C5	0.5526	0.4200	0.5153	0.4773	0.5465	0.7369	0.5393	0.6276
	C10	0.5600	0.4422	0.5177	0.4941	0.5648	0.7423	0.5580	0.6415
399,001	E	0.7493	0.4974	0.6212	0.5979	0.8506	0.3519	0.5981	0.4979
	C1	0.5372	0.4055	0.5136	0.4622	0.5455	0.0276	0.4870	0.0525
	C3	0.5450	0.3635	0.5141	0.4362	0.5849	0.0158	0.4854	0.0307
	C5	0.5471	0.3527	0.5072	0.4289	0.5679	0.0231	0.4783	0.0444
	C10	0.5539	0.3500	0.4996	0.4290	0.5366	0.0216	0.4646	0.0415
600,601	E	0.6704	0.5763	0.6628	0.6198	0.6942	0.6687	0.7015	0.6812
	C1	0.6273	0.5487	0.6201	0.5853	0.7055	0.5992	0.6819	0.6480
	C3	0.5300	0.4048	0.5301	0.4590	0.6108	0.3399	0.5682	0.4367
	C5	0.5308	0.4631	0.5312	0.4946	0.5709	0.5000	0.5661	0.5331
	C10	0.5296	0.4395	0.5373	0.4804	0.5600	0.5544	0.5712	0.5572
000,004	E	0.7380	0.5984	0.6606	0.6609	0.7932	0.5552	0.6742	0.6532
	C1	0.5026	0.3309	0.4945	0.3990	0.6667	0.0023	0.4934	0.0045
	C3	0.5023	0.3166	0.4968	0.3884	0.3976	0.0188	0.4905	0.0359
	C5	0.5053	0.3214	0.4885	0.3929	0.4936	0.0861	0.4839	0.1466
	C10	0.5262	0.3902	0.5023	0.4481	0.5169	0.1612	0.4876	0.2458
600,615	E	0.8242	0.6584	0.6901	0.7320	0.8437	0.6916	0.7194	0.7601
	C1	0.4984	0.3150	0.4727	0.3860	0.4615	0.0124	0.4727	0.0241
	C3	0.5055	0.3175	0.4850	0.3901	0.3226	0.0070	0.4774	0.0136
	C5	0.5205	0.2924	0.4800	0.3745	0.5455	0.0081	0.4684	0.0160
	C10	0.5552	0.3369	0.4919	0.4193	0.6364	0.0093	0.4576	0.0183

**Table 9 entropy-22-01162-t009:** The average metrics values of the classification results obtained by four machine learning models using the test sets.

Stock Code	Experiment Name	Average _Precision	Average _Recall	Average _Accuracy	Average _F1_Score
000,001	E	0.6820	0.7364	0.6689	0.7059
	C1	0.5417	0.6903	0.5215	0.5839
	C3	0.5314	0.6432	0.5190	0.5661
	C5	0.5471	0.6340	0.5264	0.5696
	C10	0.5623	0.6359	0.5400	0.5837
399,001	E	0.8027	0.4421	0.6189	0.5628
	C1	0.5480	0.2316	0.5027	0.2899
	C3	0.5746	0.1922	0.5024	0.2455
	C5	0.5848	0.1989	0.4998	0.2617
	C10	0.5792	0.1854	0.4876	0.2496
600,601	E	0.6797	0.6399	0.6845	0.6587
	C1	0.6622	0.5664	0.6476	0.6099
	C3	0.5654	0.4148	0.5529	0.4756
	C5	0.5430	0.5431	0.5445	0.5377
	C10	0.5288	0.5600	0.5385	0.5360
000,004	E	0.7698	0.5688	0.6669	0.6535
	C1	0.6301	0.1419	0.4930	0.1813
	C3	0.4405	0.1403	0.4916	0.1871
	C5	0.5020	0.1976	0.4881	0.2711
	C10	0.5299	0.2650	0.4975	0.3388
600,615	E	0.8343	0.6628	0.6987	0.7385
	C1	0.5159	0.1630	0.4785	0.2042
	C3	0.4804	0.1605	0.4838	0.2014
	C5	0.5663	0.1515	0.4763	0.1966
	C10	0.5698	0.1734	0.4760	0.2222

**Table 10 entropy-22-01162-t010:** The metrics values of the classification results from the test sets of LSTM and GRU.

Stock Code	Experiment Name	LSTM				GRU			
		P	R	Acc	F1	P	R	Acc	F1
000,001	E	0.7459	0.7587	0.7285	0.7523	0.7646	0.7042	0.7215	0.7231
	C1	0.5520	0.5885	0.5285	0.5697	0.5362	0.6793	0.5183	0.5993
	C3	0.5461	0.5563	0.5311	0.5512	0.5276	0.5276	0.5111	0.5276
	C5	0.5473	0.6000	0.5280	0.5724	0.5513	0.4514	0.5176	0.4963
	C10	0.6105	0.3506	0.5350	0.4454	0.5437	0.4893	0.5092	0.5150
399,001	E	0.7514	0.7716	0.7262	0.7614	0.8245	0.6447	0.7212	0.7236
	C1	0.5323	0.5391	0.5184	0.5357	0.5138	0.5881	0.5011	0.5485
	C3	0.5292	0.4615	0.5095	0.4931	0.5726	0.3637	0.5308	0.4449
	C5	0.5542	0.4930	0.5261	0.5218	0.5063	0.3419	0.4800	0.4082
	C10	0.6010	0.3489	0.5284	0.4415	0.5274	0.3800	0.4868	0.4417
600,601	E	0.7218	0.5457	0.6831	0.6215	0.6552	0.5462	0.6466	0.5958
	C1	0.4938	0.2179	0.5090	0.3024	0.4868	0.7309	0.4923	0.5844
	C3	0.4789	0.4240	0.5082	0.4498	0.4866	0.5130	0.5125	0.4994
	C5	0.5008	0.5230	0.5152	0.5116	0.4938	0.3110	0.5106	0.3817
	C10	0.4888	0.5129	0.5195	0.5006	0.4852	0.3749	0.5198	0.4230
000,004	E	0.7869	0.5557	0.6712	0.6514	0.7357	0.6307	0.6706	0.6792
	C1	0.5037	0.4208	0.4961	0.4585	0.5000	0.1442	0.4929	0.2239
	C3	0.5132	0.3999	0.5056	0.4495	0.5118	0.4347	0.5053	0.4701
	C5	0.4846	0.0615	0.4829	0.1091	0.5084	0.4908	0.4932	0.4994
	C10	0.5417	0.5709	0.5275	0.5559	0.5334	0.3196	0.5027	0.3997
600,615	E	0.7951	0.8135	0.7460	0.8042	0.8264	0.7803	0.7540	0.8027
	C1	0.5330	0.0672	0.4781	0.1194	0.5556	0.0544	0.4794	0.0991
	C3	0.5732	0.1494	0.4990	0.2370	0.5959	0.0563	0.4885	0.1028
	C5	0.5537	0.2952	0.4953	0.3851	0.6642	0.1145	0.4949	0.1954
	C10	0.7955	0.0217	0.4650	0.0422	0.5467	0.8674	0.5367	0.6707

**Table 11 entropy-22-01162-t011:** Net yield to maturity (NYR) for each strategy of the four traditional machine learning models. The data represent the cumulative rate of return at the end of the test period. The “average_NYR” represented the average value of the four traditional models.

Stock Code	Experiment Name	KNN	LOGREG	RF	SVM	Average _NYR	Buy-and-Hold
000,001	E	432.56%	268.71%	352.22%	321.92%	343.85%	69.78%
	C1	402.03%	69.23%	199.35%	155.38%	206.50%	69.78%
	C3	248.90%	167.54%	272.44%	241.41%	232.57%	69.78%
	C5	213.23%	65.05%	127.73%	321.92%	181.98%	69.78%
	C10	78.12%	67.30%	141.34%	145.34%	108.03%	69.78%
399,001	E	672.70%	415.60%	398.22%	506.70%	498.31%	112.58%
	C1	476.98%	183.13%	220.48%	10.72%	222.83%	112.58%
	C3	153.78%	47.20%	259.09%	−12.26%	111.95%	112.58%
	C5	234.79%	207.99%	−19.08%	−0.08%	105.90%	112.58%
	C10	109.64%	121.01%	182.31%	4.25%	104.30%	112.58%
600,601	E	22.25%	240.11%	273.74%	241.73%	194.46%	−25.74%
	C1	−4.80%	−12.06%	−3.24%	−44.37%	−16.12%	−25.74%
	C3	−44.10%	163.14%	−8.65%	−7.20%	25.80%	−25.74%
	C5	−25.15%	−33.82%	39.58%	122.60%	25.80%	−25.74%
	C10	−81.53%	−79.25%	123.93%	55.04%	4.55%	−25.74%
000,004	E	2236.42%	1228.62%	1816.23%	3095.86%	2094.28%	54.53%
	C1	−15.46%	11.82%	−15.41%	15.31%	−0.93%	54.53%
	C3	140.75%	−42.86%	263.30%	−36.08%	81.28%	54.53%
	C5	206.32%	146.29%	153.92%	146.77%	163.32%	54.53%
	C10	478.36%	18.60%	335.02%	137.88%	242.46%	54.53%
600,615	E	677.37%	2802.42%	1059.11%	684.73%	1305.91%	278.10%
	C1	−24.70%	−0.64%	−49.78%	−37.31%	−28.11%	278.10%
	C3	94.56%	−5.86%	10.95%	−58.79%	10.22%	278.10%
	C5	148.22%	−13.99%	−22.13%	−7.97%	26.03%	278.10%
	C10	363.71%	−53.09%	−55.83%	−13.37%	60.35%	278.10%

**Table 12 entropy-22-01162-t012:** Net yield to maturity (NYR) for each strategy of the two deep learning machine learning models. The data represent the cumulative rate of return at the end of the test period. The “average_NYR” represented the average value of two deep learning machine learning models.

Stock Code	Experiment Name	LSTM	GRU	Average _NYR
000,001	E	322.47%	380.56%	351.51%
	C1	241.83%	141.07%	191.45%
	C3	198.51%	115.63%	157.07%
	C5	188.62%	153.07%	170.84%
	C10	211.82%	57.23%	134.53%
399,001	E	629.90%	510.03%	569.96%
	C1	339.66%	234.58%	287.12%
	C3	389.41%	279.15%	334.28%
	C5	457.76%	276.86%	367.31%
	C10	373.40%	261.83%	317.62%
600,601	E	378.32%	446.18%	412.25%
	C1	77.45%	131.64%	104.54%
	C3	252.90%	147.34%	200.12%
	C5	254.26%	83.62%	168.94%
	C10	228.14%	−32.46%	97.84%
000,004	E	1580.98%	1443.52%	1512.25%
	C1	55.68%	113.66%	84.67%
	C3	334.86%	131.80%	233.33%
	C5	128.20%	388.29%	258.24%
	C10	51.22%	173.86%	112.54%
600,615	E	679.29%	1686.93%	1183.11%
	C1	51.47%	189.20%	120.33%
	C3	48.43%	183.87%	116.15%
	C5	154.64%	561.10%	357.87%
	C10	261.85%	374.65%	318.25%

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
