# Peer review of "A Labeling Method for Financial Time Series Prediction Based on Trends"

_entropy, 2020, doi:10.3390/e22101162_

Round 1
Reviewer 1 Report
see the pdf

Author Response
Please see the attachmen.

Reviewer 2 Report
I don't understand why this paper is submitted to Entropy; it has nothing to do with entropy; the word entropy is not even mentioned once
Reviewer 3 Report
In "A labeling method for financial time series prediction based on trends" the authors study an interesting and vibrant problem with applications in the analysis of observed data, in particular also in finance, where time variations in values are of course the staple of successful trading and prediction. The authors extend this to use a labeling method for financial time series prediction based on trends.
The manuscript is quite comprehensive, but the following comments should be taken into account if a revision will be granted at Entropy.
1) I would encourage the authors to extend the abstract more with the key results. As it is, the abstract is a little thin and does not quite convey the interesting results that follow in the main paper.
2) Do not use abbreviations in the abstract and elsewhere. In the absence of stringent space constraints, the use of abbreviation is not a good idea because it decreases ease of reading if a person has to remember all the abbreviations.
3) When referring to preceding research on time series prediction and in particular finance, the recent paper Clustering patterns in efficiency and the coming-of-age of the cryptocurrency market, Sci. Rep. 9, 1440 (2019) is a good reference to support this line of research. As an alternative approach, the authors should also mention wavelets, that have always been used for predicting time series. A good recent reference for predictability is Wavelet entropy-based evaluation of intrinsic predictability of time series, Chaos 30, 033117 (2020).
4) It would also improve the paper if the figure captions would be made more self contained. In addition to briefly stating what is shown, one should add a sentence or two saying what is the main message of each figure.
5) Some references contain errors, missing or incorrect information, and inconsistent formatting. The authors should correct this with the best of care.
If a revision will be granted, I will be happy to review the revised manuscript.
Round 2
Reviewer 1 Report
Thank you very much.
Author Response
Thank you for your approval.
Reviewer 2 Report
first of all, the reply is extremely strange and makes me wonder about the authors; I quote: "Author reply: I'm sorry we didn't explain "
the reply is obviously due to one author "I am sorry"; therefore I doubt about the role of other authors; in fact,one does not see any contribution from other authors in the reply
second, the references in the reply, "justifying" the submission to Entropy do not even appear in the revised version; incredible
third, suddenly, the abstract is expanded in order to explain some aim of the paper; such a text should not appear in an abstract; an abstract is an abstract ! summarizing what is found in the paper;not a set of arguments for some justification of some work; unbelievable
fourth, if one wishes to justify the work through entropy measures, why are the contributions of Miskewicz on entropy distances and correlations between time series not mentioned? like:
Miśkiewicz, J. (2010). Entropy correlation distance method. The Euro introduction effect on the Consumer Price Index. Physica A: Statistical Mechanics and its Applications, 389(8), 1677-1687. Ausloos, M., & Miśkiewicz, J. (2010). Entropy correlation distance method applied to study correlations between the gross domestic product of rich countries. International Journal of Bifurcation and Chaos, 20(02), 381-389. typos should be correctedAuthor Response
Please see the attachment.
Reviewer 3 Report
The authors have revised their manuscript comprehensively and with love to detail. I warmly recommend publication in present form.
Author Response
Thank you for your approval.